# Who Should Not Be Surveilled for HCC Development after Successful Therapy with DAAS in Advanced Chronic Hepatitis C? Results of a Long-Term Prospective Study

**DOI:** 10.3390/biomedicines11010166

**Published:** 2023-01-09

**Authors:** Alessia Ciancio, Davide Giuseppe Ribaldone, Matteo Spertino, Alessandra Risso, Debora Ferrarotti, Gian Paolo Caviglia, Patrizia Carucci, Silvia Gaia, Emanuela Rolle, Marco Sacco, Giorgio Maria Saracco

**Affiliations:** Gastro-Hepatoloy Unit, Department of Medical Sciences, University of Turin, 10126 Turin, Italy

**Keywords:** Hepatocellular Carcinoma, chronic hepatitis C, DAA, cirrhosis, FIB-4

## Abstract

**Background and aims**: The identification of patients with Hepatitis C Virus (HCV)-positive advanced chronic liver disease (aCLD) successfully treated by Direct Acting Antiviral Agents (DAAs) who really benefit from Hepatocellular Carcinoma (HCC) surveillance programs is still a matter of debate. We performed a long-term prospective cohort study on F3-F4 HCV-positive patients achieving Sustained Virologic Response (SVR) after DAAs treatment in order to identify patients who can safely suspend surveillance. **Methods:** 1000 patients with HCV-positive aCLD obtaining SVR by DAAs from January 2015 to December 2017 were divided into four groups according to baseline elastographic, ultrasonographic, clinical and biochemical features: (1) Group 1: 324 patients with Liver Stiffness Measurement (LSM) ≥ 9.5 ≤ 14.5 kPa, FIB-4 < 3.25 and APRI < 1.5 (2) Group 2: 133 patients with LSM ≥ 9.5 ≤ 14.5 kPa, FIB-4 ≥ 3.25 and/or APRI ≥ 1.5 (3) Group 3: 158 patients with LSM > 14.5 kPa, FIB-4 < 3.25 and APRI < 1.5 (4) Group 4: 385 patients with LSM > 14.5 kPa, FIB-4 ≥ 3.25 and/or APRI ≥ 1.5. FIB-4 and APRI scores were calculated at baseline and at SVR achievement. Each patient was surveiled twice-yearly by ultrasound for a median follow-up of 48 months. **Results:** among Group 1 patients, 1/324 (0.3%) developed HCC (0.09/100 patients/year [PY]), compared to 6/133 (4.5%) Group 2 patients (1.22/100 PY, *p* = 0.0009), 10/158 (6.3%) Group 3 patients (1.68/100 PY, *p* = 0.0001), 54/385 (14.0%) Group 4 patients (4.01/100 PY, *p* < 0.0001). HCC incidence was significantly lower in Group 2 compared to Group 3 (*p* = 0.004) and in Group 3 compared to Group 4 (*p* = 0.009). HCC risk fell in patients showing a decrease of FIB-4/APRI scores. **Conclusions:** the risk of HCC occurrence is negligible in about 90% of HCV-positive patients with baseline LSM ≥ 9.5 ≤ 14.5 kPa plus FIB-4 < 3.25 and APRI < 1.5 achieving SVR. Among this particular subset of patients, FIB-4/APRI scores may represent an accurate and inexpensive tool to distinguish patients not needing long-term HCC surveillance.

## 1. Lay Summary

International guidelines are divergent regarding the long-term HCC surveillance in F3 patients cured from HCV infection. The European Association for the Study of the Liver (EASL) recommends twice yearly abdominal ultrasounds in both F3 and F4 patients, while the American Association for the Study of the Liver Diseases (AASLD) recommends surveillance only in those with cirrhosis.

According to our results, the vast majority of F3 patients obtaining SVR can be safely suspended from HCC surveillance if well defined at baseline for clinical, biochemical and ultrasonographic parameters.

A not negligible absolute HCC risk persists in cirrhotic patients even if cured from HCV infection, and more data are needed before establishing definitive algorithms able to drive cost-effective strategies of HCC surveillance on an individual basis.

## 2. Introduction

The clearance of Hepatitis C Virus (HCV) by Direct Acting Antiviral Agents (DAAs) reduces but does not eliminate the risk of developing Hepatocellular Carcinoma (HCC) in patients with advanced chronic liver disease (aCLD) [1]. For this reason, there is general agreement that cirrhotic patients should undergo HCC surveillance after achieving Sustained Virologic Response (SVR), but the international guidelines are divergent regarding the follow-up of F3 patients. According to AASLD guidelines [2], F3 patients obtaining SVR may be safely discharged, while the European guidelines [3] are more stringent, recommending ultrasonography every 6 months in both F3 and F4 patients, irrespective of the results of Non Invasive Tests (NITs) performed before and after SVR achievement [4]. This discrepancy is mainly based upon the difficulty in precisely defining patients with F3 fibrosis; moreover, the histologic stage is not the only predictor of HCC and for this reason there is currently a general consensus that attribution to the F3 stage should depend not only on histologic/elastographic aspects but also on ultrasonographic, clinical and biochemical variables [1,5] in order to accurately predict the HCC risk on an individual basis. To date, several studies [1,6,7,8] have tried to define the most effective NITs and related algorithms for identifying who to follow over time, while few studies with adequate follow-up [5,9,10,11] have addressed the problem of sustained responders to DAAs with baseline pre-cirrhotic liver fibrosis who can be safely suspended from long-term follow-up. This issue is relevant, as its definition would avoid unnecessary examinations and visits, freeing up resources for patients at high risk of developing HCC.

The primary aim of our study was to establish the long-term risk of HCC in patients with aCLD successfully treated by DAAs stratified according to their baseline measurement of liver stiffness (LSM), liver ultrasonographic aspect, clinical history, and Fibrosis-4 Index for Liver Fibrosis (FIB-4)/AST to Platelet Ratio Index (APRI) values. The secondary aim was to verify whether a post-SVR change in FIB-4/APRI values adds relevant information regarding patients who may terminate long-term HCC surveillance.

## 3. Patients and Methods

Due to Italian National Health System rules, since the introduction of DAAs in Italy from 2015 to 2017, only patients with advanced liver fibrosis (F3–F4 according to METAVIR Score [12]) could be reimbursed for treatment with DAAs; for this reason, all consecutive patients with HCV-positive aCLD referred to the Gastrohepatologic Clinic of Molinette Hospital, Turin, Italy, for DAAs therapy (sofosbuvir-based 87%) according to the EASL guidelines [13] between 1 January 2015 and 31 December 2017 were considered.

Inclusion criteria were as follows: >18 years; positive HCV-RNA by polymerase chain reaction (PCR); F3 and F4 liver fibrosis assessed by Transient Elastography (TE) within 3 months prior to inclusion into the study. Exclusion criteria were lack of written informed consent, TE not feasible or unreliable measurement, patients on waiting list for orthotopic liver transplant (OLT), post-OLT patients, past or current history of HCC, concomitant liver diseases such as haemochromatosis, Wilson’s disease, drug-related liver disease, autoimmune hepatitis, HBsAg carriership, Human immunodeficiency Virus (HIV) infection, primary biliary cholangitis, and alpha-1-antitrypsin deficiency.

Patients were considered cirrhotic if showing LSM > 14.5 kPa by TE; stage 3 liver fibrosis was established by LSM values ranging from 9.5 to 14.5 kPa [14,15], but patients with a nodular liver surface, splenomegaly, porto-systemic collaterals detected by abdominal ultrasound (US), platelets < 120 × 10^9^/L, previous or current history of ascites, encephalopathy and variceal bleeding, or esophageal varices detected by endoscopy were considered cirrhotic [16], irrespective of their baseline LSM value. The reliability criteria for LSM were as follows: 10 valid measurements achieved with a success number ≥60% and an interquartile range-to-median ratio ≤30%. A minimum of 3 h fasting was required.

Out of 1188 consecutive patients treated with DAAs, 146 (12.2%) were excluded (HBV and/or HIV carriers, past or current history of HCC, pre/post-OLT patients, TE not feasible, lack of informed consent). Of the remaining 1042 patients, 23 (2.2%) did not achieve SVR (negative viremia 12 weeks after the end of treatment) and were excluded from the study. Out of 1019 sustained responders, 19 (1.9%) were lost to follow up. Therefore, the final analysis was performed in 1000 patients with SVR. The patient flow is reported in Figure 1.

At baseline, a complete medical history and physical examination was undertaken and the following data were obtained from each patient: age, gender, ethnicity, smoking habits, alcohol intake, body mass index (BMI), waist circumference (WC), duration of HCV infection, and relevant co-morbidities. The following data on laboratory parameters were also recorded to define baseline characteristics: complete blood count, routine liver biochemistry (alanine aminotransferase [ALT] and AST, total bilirubin, albumin, alkaline phosphatase [APH], gamma glutamyltranspeptidase [GGT]), international normalized ratio [INR], creatinine, fasting plasma glucose [FPG], total cholesterol and HDL, triglycerides, HCV genotype, and viral load [AmpliPrep^®^/COBAS Taqman^®^ HCV test, Roche Diagnostics, Basel, Switzerland]. We adopted an algorithm based on BMI, waist circumference, triglycerides and GGT to detect and graduate baseline liver steatosis by the fatty liver index [17]: a significant liver steatosis was associated with patients showing a cut-off ≥ 60.

For each patient, we calculated FIB-4 [(Age (years)) × AST (U/L)/(Platelets (10^9^/L)\1 × \2√ALT (U/L))] [18] and APRI [(AST/AST upper limit normal)/Platelets (10^9^/L) × 100] [19] scores at baseline and at SVR achievement. An abdominal ultrasound (US) was performed on each patient at baseline in order to exclude HCC. Cirrhotic patients were stratified according to the Child-Turcotte-Pugh classification and Model for End-Stage Liver Disease [MELD] and underwent endoscopy to verify the presence of esophageal varices.

Patients were divided into 4 groups according to their baseline characteristics:(1)Group 1: patients with LSM ≥ 9.5 ≤ 14.5 kPa and FIB-4 < 3.25 and APRI < 1.5(2)Group 2: patients with LSM ≥ 9.5 ≤ 14.5 kPa and FIB-4 ≥ 3.25 or APRI ≥ 1.5(3)Group 3: patients with LSM > 14.5 kPa or clinical/biochemical/US signs of cirrhosis and FIB-4 < 3.25 and APRI < 1.5(4)Group 4: patients with LSM > 14.5 kPa or clinical/biochemical/US signs of cirrhosis and FIB-4 ≥ 3.25 or APRI ≥ 1.5

We used the FIB-4 cut-off of ≥ 3.25 due to its positive predictive value of 82.1% for advanced fibrosis [20]; in order to better classify patients showing an indeterminate range (1.45–3.25), we used the APRI cut-off ≥ 1.5 to individuate carriers of significant fibrosis [21].

An abdominal US was planned at 6-month intervals and patients were followed until OLT, death or until the end of December 2021. Patients were also censored at the moment of HCC diagnosis, performed according to international guidelines [22,23] by histological examination or by contrast-enhanced imaging methods showing hypervascularity in late arterial phase and washout on portal venous and/or delayed phases. They were followed-up for a median of 48 (IQR: 36–60) months after achieving SVR.

The study was performed in accordance with the principles of the Declaration of Helsinki and approved by the local ethics committee (Comitato Etico Interaziendale Città della Salute e della Scienza di Torino, Turin, Italy, n 452), and written informed consent was obtained from all patients.

### Statistical Analysis

Continuous variables were reported as median (inter-quartile range (IQR)). Normality was checked by the D’Agostino–Pearson test. Categorical variables were reported as number and percentage. Comparison of continuous variables between independent groups was performed by the Mann–Whitney test. Regarding the dichotomous categorical variable, a Chi-squared test was performed for unpaired analysis. Comparison of Kaplan-Meier survival curves was performed using the Logrank test. The association between variables was assessed by Cox proportional hazards, the strength of association was reported as hazards ratio (H.R.) and 95% CI, respectively.

All statistical analyses were performed using MedCalc^®^ v.18.9.1 (MedCalc Software Ltd., Ostend, Belgium), and a *p* value ≤ 0.05 was considered statistically significant.

## 4. Results

Baseline characteristics of the patients divided into four groups are reported in Table 1 and Table 2.

Group 1 patients showed lower median LSM (11.6 kPa [IQR: 10.4–12.8]) compared with the other groups (*p* = 0.0001), a shorter duration of infection (median 17 years, [IQR: 10–23], *p* = 0.01), fewer baseline co-morbidities (20.7%, *p* = 0.005), a lower rate of diabetes mellitus (10.8%, *p* = 0.0002), and a significantly lower fatty liver index (37.2 [IQR: 21.4–57.9], *p* = 0.0001). Markers of liver synthesis were significantly higher in group 1 patients (median albumin value: 4.4 g/dL [IQR:4.2–4.6], *p* = 0.007, median total cholesterol level: 157.5 mg/dL [IQR: 130.5–182.5], *p* = 0.005) as well as the median platelet count (187.5 × 10^3^/mm^3^ [IQR: 158.5–231.0], *p* = 0.0002), confirming a less advanced liver disease (median MELD: 7.0 [IQR: 6.0–7.0], *p* = 0.0001) and a lower stage of liver fibrosis (median FIB-4 score: 1.8 [IQR: 1.3–2.4], *p* = 0.0001, median APRI score: 0.3 [IQR: 0.2–0.4], *p* = 0.003). Overall, there were 71 incident cases of HCC diagnosed during follow-up, with an HCC incidence rate (IR) of 1.97/100 patients/year (PY) and a cumulative incidence rate (CIR) of 2.2%, 5.5% and 9.5% at 12, 36 and 60 months, respectively. In Group 1, one out of 324 patients (0.3%) developed HCC 7 months after SVR achievement, corresponding to an HCC IR of 0.09/100 PY and a CIR at 12, 36 and 60 months of 0.3%, compared with 6/133 (4.5%) in Group 2 (1.22/100 PY, CIR of 2.3%, 3.3%, 5.7%), 10/158 (6.3%) in Group 3 (1.68/100 PY, CIR of 0.7%, 4.2%, 8.6%) and 54/385 (14.0%) in Group 4 (4.01/100 PY, CIR of 4.3%, 11.3%, 17.8%). The difference in HCC incidence between Group 1 and the other three Groups was statistically significant (*p* = 0.0009 vs. Group 2, *p* = 0.0001 vs. Group 3, *p* < 0.0001 vs. Group 4); a significant lower HCC IR was also observed in Group 2 compared with Group 4 (*p* = 0.004) and in Group 3 vs. Group 4 (*p* = 0.009). (Figure 2). In order to individuate independent predictors of HCC occurrence, we checked patients for each baseline demographic, clinical and biochemical characteristic reported in Table 1 and Table 2; Table 3 presents the factors associated with the long-term risk of HCC by univariate and multivariate analysis. Pre-therapy variables which were not significant in the models were not reported. In particular, the prevalence of various genotypes did not significantly change among patients who developed (or did not) HCC (genotype 1 = 71.7% vs. 69.5%, genotype 2 = 1.6% vs. 4.6%, genotype 3 = 8.4% vs. 8.6%, genotype 4 = 18.3% vs. 11%, *p* = 0.43). According to the univariate analysis, many characteristics were associated with HCC occurrence (older age, male gender, high LSM, cirrhosis, presence of esophageal varices, APRI/FIB-4/MELD scores, low levels of albumin, cholesterol and platelet count, belonging to group 2–4) but only male gender (HR = 1.93; 95% CI = 1.12–3.31, *p* = 0.02), cirrhosis (HR = 3.6; CI = 2.42–5.88, *p* = 0.001), the presence of esophageal varices (HR = 1.76; CI = 1.03–3.01, *p* = 0.04), and belonging to group 2–4 (HR = 10.95; CI = 1.19–100.51, *p* = 0.03) were independent predictors of HCC. At the end of the follow-up, there was an overall mortality of 60 out of 1000 patients (6%): 7/324 (2.1%) in group 1, 6/133 (4.5%) in group 2, 12/158 (7.6%) in group 3, and 35/385 (9.1%) in Group 4 (Figure 3). Death occurred due to liver-related causes in 42 patients (70%): one out of 42 (2.4%) belonged to Group 1 and died for the HCC occurrence as well as 3 Group 2 patients (7.1%). The remaining 18 patients died due to extrahepatic neoplastic causes or cardiovascular diseases. Ten patients (1.0%) were referred for liver transplantation, 2 of them (20%) belonging to Group 2 and 8 (80%) to Group 4. The difference in terms of mortality was statistically significant between Group 1 and Group 3 (*p* = 0.007) and between Group 1 and Group 4 (*p* = 0.0001).

### Impact of Change in FIB-4/APRI on HCC Risk

Due to the very small proportion (0.8%) of patients showing an increase in FIB-4/APRI scores after achieving SVR, we focused on Group 2 and 4 patients and we observed that the risk of incident HCC was reduced among patients who experienced a decrease of FIB-4/APRI scores below 3.25 and 1.5, respectively. In Group 2, two out of 85 (2.4%) patients showing a decline in FIB-4/APRI developed HCC compared with 4/48 (8.3%) patients with scores above the cut-offs, corresponding to an IR of 0.6/100 PY vs. 2.8/100 PY and a CIR at 12, 36 and 60 months of 1,2%, 2.6% and 2.6%, respectively, compared to 4.7%, 4.7% and 14.3%, *p* = 0.049 (Figure 4), respectively. In Group 4, 12/167 (7.2%) patients with a significant decrease in FIB-4/APRI scores showed “de novo” HCC compared to 42/218 (19.3%) patients with no relevant changes; HCC IR was 1.7/100 PY in the first sub-group versus 6.4/100 PY in the second one, with a CIR of 0.6%, 5.8% and 8.1% compared with 7.2%, 15.9%, 26.8%, respectively, *p* < 0.0001 (Figure 5). Baseline characteristics of patients stratified according to their improvement in FIB-4/APRI scores are reported in Table 4; the only independent predictor of HCC by multivariate analysis was the lack of FIB-4/APRI improvement (H.R. = 0.02 [95% CI: 0.004–0.15]).

## 5. Discussion

The current International Guidelines [2,3] diverge on the surveillance strategy regarding patients with aCLD achieving SVR, in particular those ones with baseline F3 fibrosis. The EASL [3] recommends long-term US surveillance in F3 patients, whereas AASLD does not [2]. This divergence is mainly based upon the difficulty in establishing an accurate diagnosis of F3 fibrosis: liver biopsy can underestimate the presence of cirrhosis in patients histologically labeled as F3 [24], while TE is unable to distinguish F3 fibrosis from cirrhosis with certainty [25]. In order to better stratify the neoplastic risk in this particular subset of patients, recent studies [9,10,11] have tried to associate algorithms composed by demographic, clinical, biochemical risk factors to the histological or elastographic assessment, but to date none of them have been officially recommended.

According to our data, the HCC incidence in patients with LSM ≥ 9.5 ≤ 14.5 kPa achieving SVR is negligible (0.09/100 PY) in the vast majority of them after a median follow-up of 4 years; the identification of this low-risk group can be performed easily before treatment by associating clinical, biochemical variables and US imaging to LSM. Moreover, a significant decline in FIB-4/APRI scores after SVR achievement in this particular subset of patients with high baseline values is associated with a decreased risk of HCC (0.6/100 PY), enabling the individuation of additional patients at low risk of HCC. Overall, 90% of patients with LSM ≥ 9.5 ≤ 14.5 kPa were identified as having an HCC risk < 1%/year, well below the cost-effectiveness cut-off for HCC surveillance. On the other hand, the remaining 10% of patients with persistently high FIB-4/APRI scores were probably carriers of undiagnosed cirrhosis, therefore needing an HCC surveillance program due to the relevant HCC incidence found (2.8/100 PY).

Our results are similar to those reported by recent studies [5,9,10,11], which specifically addressed the issue of HCC risk in pre-cirrhotic patients. By adopting algorithms based upon FIB-4/APRI scores, Kanwal et al. [10] showed that only 4% of patients without a diagnosis of cirrhosis at baseline require HCC surveillance. Sanchez-Azofra et al. [5] followed-up a large cohort of patients with baseline HCV-positive stage-3 fibrosis successfully treated by DAAs, finding a low HCC incidence rate (0.47/100 PY), suggesting the maintenance of surveillance only in males older than 55 years of age. According to our data, male sex is an independent predictor of HCC, as well as cirrhosis, persistently high FIB-4/APRI scores, and clinical significant portal hypertension, confirming what was reported by Azzi et al. [9]. Surprisingly, baseline co-morbidities and variables such as diabetes, metabolic syndrome, high BMI, visceral obesity, and fatty liver index were not selected as independent HCC risk predictors, but this finding is consistent with data from previous studies [5,9,10,11]. In particular, diabetes mellitus did not emerge as an unfavourable independent predictive factor for HCC risk upon multivariate analysis. Data from follow-up studies on patients with SVR after therapy with DAAs [9,10,11,26,27,28,29,30,31,32,33,34,35,36,37,38] are conflicting, as diabetes was not found to be an independent predictor of HCC in most of them [9,10,11,26,27,28,30,31,32,33,34,35].

Our results confirmed what was already found by Kanwal et al. [10], who showed that the HCC risk remained above the accepted threshold for surveillance in cirrhotic patients, even in the case of significant decline in FIB-4/APRI scores after therapy. However, at variance with our data, a recent study [11] showed that a post-treatment algorithm combining age/albumin/TE and, optionally, alfafetoprotein and alcohol intake enables a reliable risk stratification for the development of HCC, even among cirrhotic patients, identifying a large proportion of patients with an HCC risk < 1%/year, below the threshold considered to be cost-effective. Cost-effectiveness analyses regarding HCC surveillance changed over time; Cucchetti et al. [39] quoted an annual HCC risk threshold > 1.5%, above which HCC screening was cost-effective, but recent studies [40,41] focusing on patients achieving SVR suggested lower thresholds, from 1.32/100 PY up to 0.5/100 PY. According to Mueller et al. [42], HCC surveillance is cost-effective until the age of 70 for cirrhotic patients cured of hepatitis C and until the age of 60 for patients with non-cirrhotic advanced fibrosis. Irrespective of the thresholds considered, we acknowledge that FIB-4/APRI-based algorithms may not be the best method to stratify the HCC risk in cirrhotic patients achieving SVR; we have recently compared [43] different non-invasive scoring systems (the Forns index, FIB-4, albumin-bilirubin score [ALBI], age/gender/albumin-bilirubin/platelets score [aMAP]) in cirrhotics successfully treated by DAAs, and we found that ALBI showed the highest diagnostic accuracy for the detection of HCC, confirming previous data [44,45,46]. However, given the controversies regarding the markers and algorithms which should be adopted, we think that results from large prospective studies with longer follow-up are still needed before incorporating definite algorithms into official recommendations regarding the optimal HCC surveillance strategy in a cirrhotic population cured from HCV infection.

Very similar to what was previously reported [47], death occurred in about 6% of our patients due to liver-related causes in the majority of them (70%); however, the mortality rate was significantly lower among Group 1 patients compared with Group 3–4 patients, confirming the excellent clinical outcome of F3 patients with low FIB-4/APRI scores obtaining SVR. The strengths of our study are the prospective design, the large sample size with long-term follow-up, and the very low rate of patients lost to follow-up. However, our study was conducted in a single academic centre mainly recruiting Caucasian patients without a validation cohort, therefore limiting the reproducibility of our conclusions, and multicenter prospective studies are needed in order to make a solid recommendation.

Moreover, we acknowledge the possibility that some cirrhotic patients could have been included in Group 1 and 2, due to the wide range of kPa adopted to define F3 patients and despite the accurate clinical and ultrasonographic screening, however, according to our results, our approach would still work for these patients.

In conclusion, the great majority of patients with LSM ≥ 9.5 ≤ 14.5 kPa achieving SVR can be safely suspended from HCC surveillance if well defined at baseline for clinical, biochemical and US parameters, which are all currently used, before starting antiviral therapy with no cost increases. On the other hand, a word of caution is still necessary regarding the withdrawal or deferral of HCC surveillance in cirrhotic patients cured from HCV infection, as a late diagnosis of HCC implies severe consequences for an individual patient.

## Figures and Tables

**Figure 1 biomedicines-11-00166-f001:**
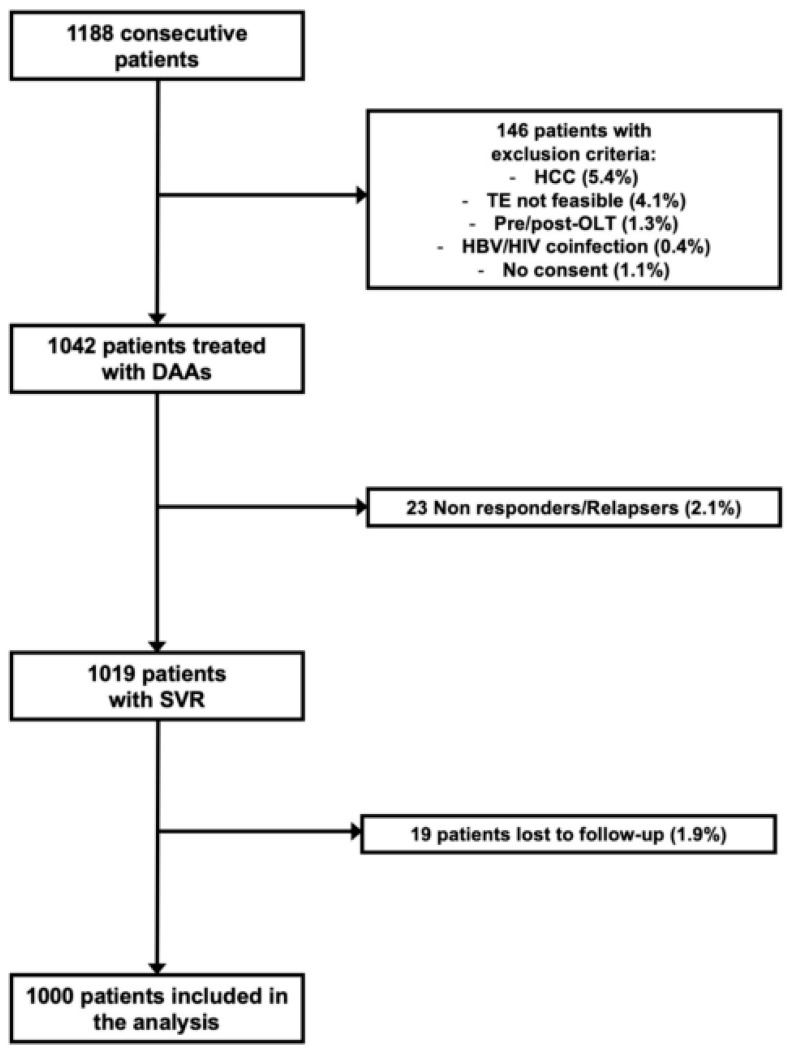
Flowchart of recruited patients.

**Figure 2 biomedicines-11-00166-f002:**
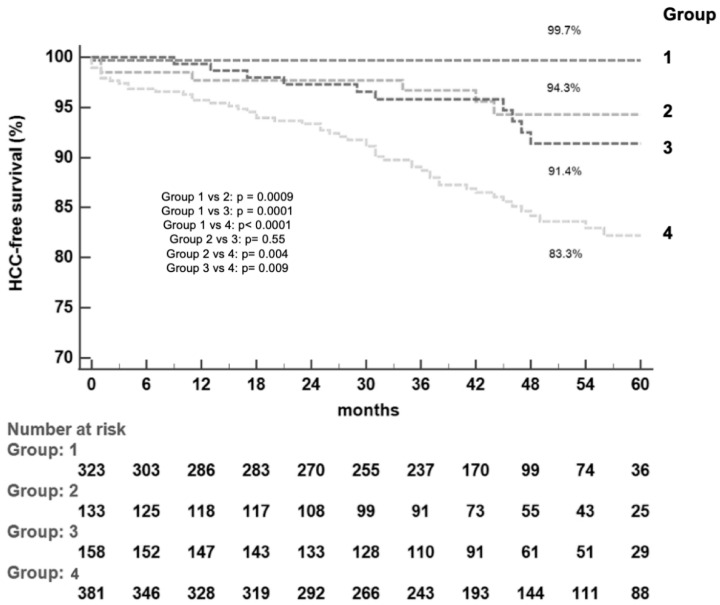
Kaplan-Meier HCC-free survival curves of patients according to group.

**Figure 3 biomedicines-11-00166-f003:**
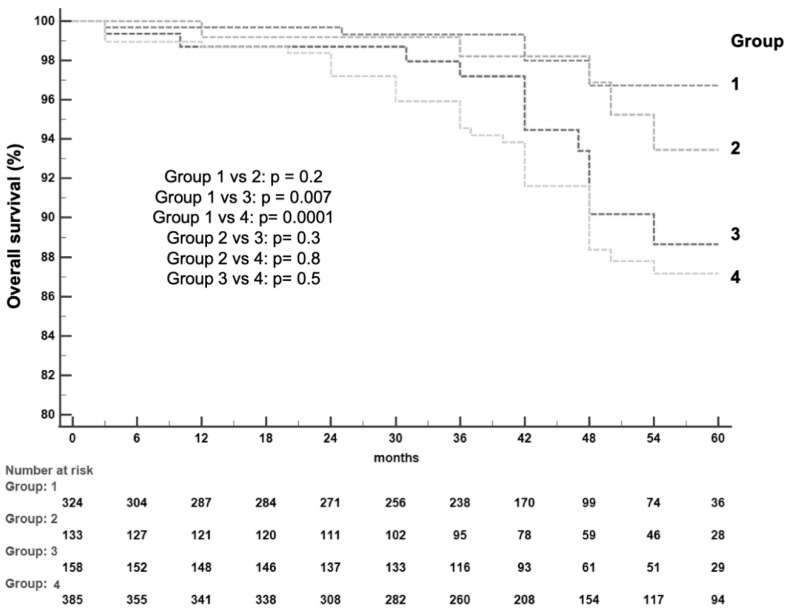
Kaplan-Meier overall-survival curves of patients according to group.

**Figure 4 biomedicines-11-00166-f004:**
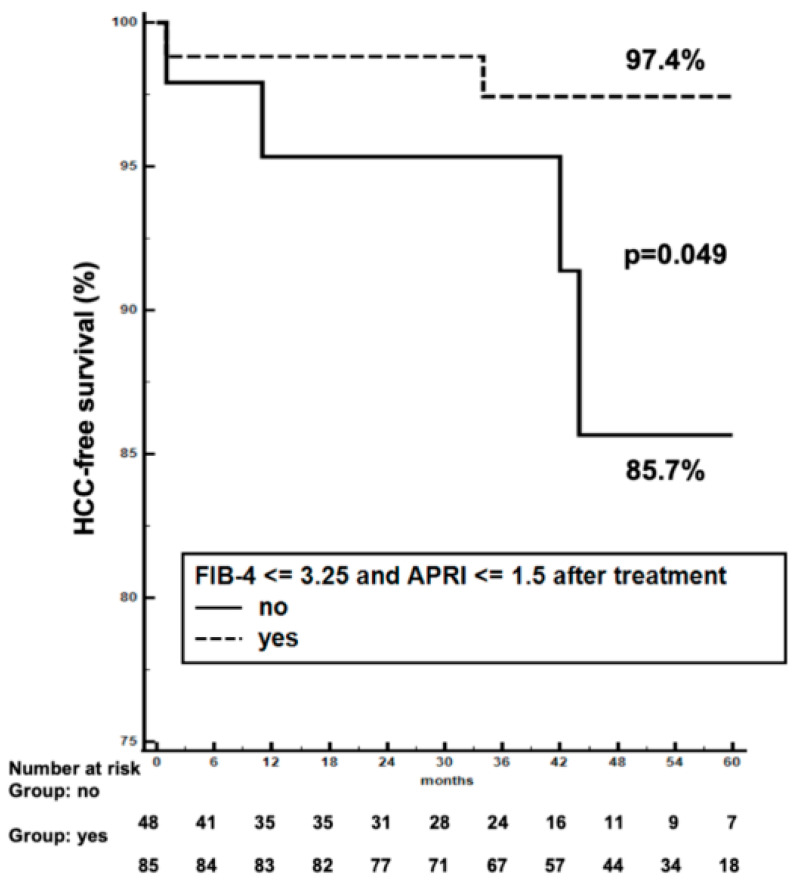
Kaplan-Meier HCC-free survival curves of Group 2 patients according to changes in FIB-4/APRI scores.

**Figure 5 biomedicines-11-00166-f005:**
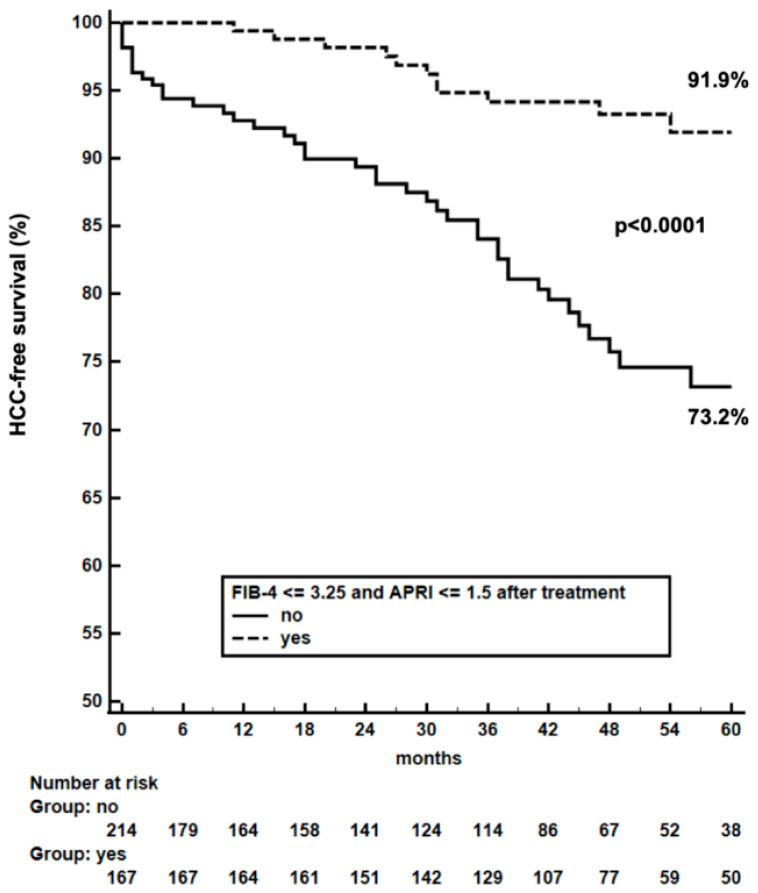
Kaplan-Meier HCC-free survival curves in Group 4 patients according to changes in FIB-4/APRI scores.

**Table 1 biomedicines-11-00166-t001:** Baseline demographic and clinical characteristics of 1000 sustained responders to DAAs.

Characteristics	Group 1 *n* = 324 (32.4%)	Group 2 *n* = 133 (13.3%)	Group 3 *n* = 158 (15.8%)	Group 4 *n* = 385 (38.5%)	*p*
Age (years), median (IQR)	56 (50–65)	68 (60–75)	53 (48–64)	65 (55–74)	Group 1 vs. 2/4 = 0.0001 Group 2 vs. 3/4 = 0.002 Group 3 vs. 4 = 0.0001
Males, *n* (%)	186 (57.4%)	60 (45.1%)	112 (70.9%)	221 (57.4%)	Group 1 vs. 2/3 = 0.002 Group 2 vs. 3/4 = 0.01 Group 3 vs. 4 = 0.003
Females, *n* (%)	138 (42.6%)	73 (54.5%)	46 (29.1%)	164 (42.6%)	
Ethnicity					N.S.
Caucasian, *n* (%)	316 (97.5%)	133 (100%)	155 (98.1%)	381 (99%)	
African, *n* (%)	8 (2.5%)	0 (0%)	3 (1.9%)	4 (1%)	
BMI, median (IQR)	24.8 (22.6–27.4)	24.2 (22.1–27.8)	25 (22.8–28.4)	24.7 (22.7–27.1)	N.S.
BMI > 25, *n* (%)	153 (47.2%)	54 (40.6%)	79 (50%)	178 (46.2%)	N.S.
Abnormal waist circumference, *n* (%)	178 (54.9%)	94 (70.6%)	86 (54.4%)	253 (65.7%)	Group 1 vs. 2/4 = 0.002 Group 2 vs. 3 = 0.005 Group 3 vs. 4 = 0.01
Obesity, *n* (%)	35 (10.8%)	18 (13.5%)	19 (12.0%)	38 (9.9%)	N.S.
Duration of infection (years), median (IQR)	17 (10–23)	19 (11–24)	18 (7–24)	20 (11–24)	Group 1 vs. 4 = 0.01
Smoking status					N.S.
Never, *n* (%)	291 (89.8%)	121 (91%)	130 (82.3 %)	345 (89.6%)	
Past or current, *n* (%)	33 (10.2%)	12 (9%)	28 (17.7%)	40 (10.4%)	
Alcohol intake					N.S.
No, *n* (%)	315 (97.2%)	132 (99.2%)	149 (94.3%)	375 (97.4%)	
Yes, *n* (%)	9 (2.8%)	1 (0.8%)	9 (5.7%)	10 (2.6%)	
Diabetes, *n* (%)	35 (10.8%)	23 (17.2%)	35 (22.2%)	82 (21.3%)	Group 1 vs. 3/4 = 0.0002
Metabolic syndrome, *n* (%)	47 (14.5%)	24 (18%)	32 (20.3%)	73 (19.0%)	N.S.
Baseline co-morbidities, *n* (%)	67 (20.7%)	48 (36.1%)	44 (27.8%)	115 (29.9%)	Group 1 vs. 2/4 = 0.005
Fatty liver index, median (IQR)	37.2 (21.4–57.9)	46.2 (26.8–63.8)	48.8 (28.3–73.0)	47.6 (30.5–68.0)	Group 1 vs. 2/3/4 = 0.0001
Fatty liver index ≥ 60, *n* (%)	77 (23.8%)	37 (27.8%)	65 (41.1%)	135 (35.1%)	Group 1 vs. 3/4 = 0.001 Group 2 vs. 3 = 0.02
Esophageal varices, *n* (%)	0 (0%)	0 (0%)	26 (16.5%)	178 (46.2%)	Group 1/2 vs. 3/4 = 0.0001

DAAs, Direct Antiviral Agents; IQR, Interquartile range; BMI, Body Mass Index; N.S., Not Significant.

**Table 2 biomedicines-11-00166-t002:** Baseline biochemical, virologic and liver-related characteristics of 1000 sustained responders to DAAs.

Characteristics	Group 1 *n* = 324 (32.4%)	Group 2 *n* = 133 (13.3%)	Group 3 *n* = 158 (15.8%)	Group 4 *n* = 385 (38.5%)	*p*
Liver stiffness (kPa), median (IQR)	11.6 (10.4–12.8)	12.6 (11.4–14.0)	18.2 (16.0–23.7)	23.4 (16.9–32.4)	Group 1 vs. 2/3/4 = 0.0001 Group 2 vs. 3 = 0.0001 Group 3 vs. 4 = 0.0001
FIB-4 score, median (IQR)	1.8 (1.3–2.4)	4.5 (3.7–6.1)	2.1 (1.5–2.7)	6.2 (4.5–9.3)	Group 1 vs. 2/3/4 = 0.0001 Group 2 vs. 3/4 = 0.003 Group 3 vs. 4= 0.0001
APRI score, median (IQR)	0.3 (0.2–0.4)	0.7 (0.4–1.1)	0.3 (0.2–0.4)	0.9 (0.6–1.4)	Group 1 vs. 2/3/4 = 0.003 Group 2 vs. 3/4 = 0.001 Group 3 vs. 4 = 0.0001
MELD score, median (IQR)	7 (6–7)	7 (7–9)	7 (7–8)	8 (7–10)	Group 1 vs. 2/3/4 = 0.003 Group 2 vs. 3/4 = 0.03 Group 3 vs. 4= 0.0001
MELD > 15, *n* (%)	0 (0%)	0 (0%)	8 (5.1%)	16 (4.2%)	Group 1 vs. 3/4 = 0.0001 Group 2 vs. 3/4 = 0.0001
AST (IU/mL), median (IQR)	47 (33.5–57.5)	80 (50–121.8)	50 (33–68)	84 (57–117)	Group 1 vs. 2/4 = 0.0001 Group 2 vs. 3 = 0.001 Group 3 vs. 4 = 0.0001
ALT (IU/mL), median (IQR)	64 (41–87)	79 (51.8–140.5)	64.5 (38–88)	72 (51–121)	Group 1 vs. 2/4 = 0.0001 Group 2 vs. 3 = 0.0001 Group 3 vs. 4 = 0.0002
GGT (IU/mL), median (IQR)	31 (28–71)	60 (37.8–102.3)	64 (51–105)	64 (48–103)	Group 1 vs. 2/3/4 = 0.0008 Group 2 vs. 4 = 0.02
Platelets count (×10^3^/mm^3^), median (IQR)	188 (159–231)	123 (98–150)	163 (133–213)	92 (66–123)	Group 1 vs. 2/3/4 = 0.0002 Group 2 vs. 3/4 = 0.0001 Group 3 vs. 4 = 0.0001
Albumin (g/dL), median (IQR)	4.4 (4.2–4.6)	4.2 (3.9–4.4)	4.3 (4–4.5)	3.9 (3.5–4.2)	Group 1 vs. 2/3/4 = 0.007 Group 2 vs. 3/4 = 0.0007 Group 3 vs. 4 = 0.0001
Total cholesterol (mg/dL), median (IQR)	158 (131–183)	149 (130–170)	144 (130 - 165)	130 (123–156)	Group 1 vs. 2/3/4 = 0.005 Group 2 vs. 4 = 0.0003 Group 3 vs. 4 = 0.0001
Triglycerides (mg/dL), median (IQR)	98 (72–116)	96 (75–109)	96 (68–110)	100 (77–107)	N.S.
Genotypes, 1	236 (72.8%)	106 (79.7%)	113 (71.5%)	290 (75.3%)	N.S.
2	15 (4.6%)	8 (6.0%)	6 (3.8%)	18 (4.7%)	
3	35 (10.8%)	6 (4.5%)	15 (9.5%)	30 (7.8%)	
4	38 (11.8%)	13 (9.8%)	24 (15.2%)	47 (12.2%)	

DAAs, Direct Antiviral Agents; IQR, Interquartile range; N.S., Not Significant.

**Table 3 biomedicines-11-00166-t003:** Association between baseline characteristics and development of HCC.

	Hazard Ratio (95% CI)			
Characteristic	Univariate	*p*	Multivariate	*p*
Age in years	1.02 (1.00–1.05)	0.03	1.02 (0.996–1.043)	0.10
Male gender	1.75 (1.05–2.91)	0.03	1.93 (1.12–3.31)	0.02
Liver stiffness (kPa)	1.04 (1.03–1.06)	<0.0001	1.01 (0.997–1.030)	0.12
Cirrhosis (yes)	4.02 (2.81–6.43)	0.0001	3.6 (2.42–5.88)	0.001
APRI > 1.5	3.42 (2.04–5.75)	<0.0001	1.36 (0.76–2.44)	0.31
FIB-4 > 3.25	4.34 (2.38–7.99)	<0.0001	0.82 (0.35–1.92)	0.65
MELD > 15	4.01 (1.61–9.58)	0.003	2.3 (0.88–6.00)	0.09
Esophageal varices (yes)	4.52 (2.83–7.20)	<0.0001	1.76 (1.03–3.01)	0.04
Platelet count < 120.000/mm^3^	4.28 (2.55–7.18)	<0.0001	1.54 (0.75–3.18)	0.24
Albumin < 4 g/dL	1.68 (1.55–1.75)	<0.0001	1.37 (0.97–1.61)	0.06
Total cholesterol < 150 mg/dL	1.01 (1.002–1.16)	0.04	1.06 (0.998–1.014)	0.17
Group 2 (reference Group 1)	14.49 (1.74–120.34)	0.01	10.95 (1.19–100.51)	0.03
Group 3–4 (reference Group 1)	38.83 (5.39–279.91)	0.0003	15.8 (2.03–123.44)	0.009

HCC, Hepatocellular Carcinoma; CI, confidence interval.

**Table 4 biomedicines-11-00166-t004:** Baseline characteristics of Group 2 and 4 patients according to their improvement in FIB-4/APRI scores.

	Improved (252 pts)	Not Improved (266 pts)	*p*
Age (years), median [IQR]	66 [57–74]	66.5 [56–74]	0.89
Males (*n*, %)	124 (49.2%)	157 (59%)	0.03
Females (*n*, %)	128 (50.8%)	109 (41%)	
Ethnicity			0.051
Caucasian (*n*, %)	252 (100%)	262 (98.5%)	
African (*n*, %)	0 (0%)	4 (1.5%)	
BMI, median (IQR)	24.1 (22.1–26.9)	25.3 (22.9–27.6)	0.004
BMI > 25, *n* (%)	158 (37.3%)	138 (51.9%)	0.0009
Abnormal waist circumference, *n* (%)	177 (70.2%)	170 (63.9%)	0.62
Obese patients (*n*, %)	26 (10.3%)	30 (11.3%)	0.73
Smoking status			0.16
Never (*n*, %)	230 (91.3%)	236 (88.7%)	
Past or current smokers (*n*, %)	22 (8.7%)	30 (11.3%)	
Alcohol abuse			0.32
No (*n*, %)	245 (97.2%)	262 (98.5%)	
Yes (*n*, %)	7 (2.8%)	4 (1.5%)	
Infection duration (years), median (IQR)	20 (11–24)	19 (13–24)	0.76
Metabolic syndrome (*n*, %)	57 (22.6%)	40 (15%)	0.03
Baseline co-morbidities (*n*, %)	78 (31%)	85 (32%)	0.81
Liver stiffness (kPa), median (IQR)	17.3 (14–26.8)	20.6 (14–30.7)	0.009
FIB-4, median (IQR)	4.5 (3.8–5.8)	7.7 (4.0–11.5)	<0.0001
APRI, median (IQR)	0.7 (0.5–1)	1 (0.7–1.6)	<0.0001
MELD > 15, *n* (%)	3 (1.2%)	13 (4.9%)	0.02
Esophageal varices, *n* (%)	51 (20.2%)	127 (47.7%)	<0.0001
ALT (IU/L), median (IQR)	81 (55.5–140)	70 (48–109)	0.0004
GGT (IU/L), median (IQR)	62 (45–109)	61.5 (46–100)	0.6
Platelet count (×10^3^/mm^3^), median (IQR)	120 (99–145)	80.5 (57–106)	<0.0001
Albumin (g/dL), median (IQR)	4.2 (3.9–4.4)	3.8 (3.4–4.1)	<0.0001
Total cholesterol (mg/dL), median (IQR)	144 (130–165.5)	130 (120–155)	0.0004
FLI, median (IQR)	46.6 (28–68.3)	47.2 (30.3–67.4)	0.72

BMI, body mass index; MELD, Model of End Stage Liver Disease; IQR, Inter Quartile Range; ALT, alanine aminotransferase; GGT, gamma glutamyl transpeptidase; FLI, Fatty Liver Index.

## Data Availability

The data that support the findings of this study are available from the corresponding author upon reasonable request.

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
