# Peer review of "Who Should Not Be Surveilled for HCC Development after Successful Therapy with DAAS in Advanced Chronic Hepatitis C? Results of a Long-Term Prospective Study"

_biomedicines, 2023, doi:10.3390/biomedicines11010166_

Round 1
Reviewer 1 Report
It is a prospective study that brings new data about the usefulness of hepatocellular carcinoma (HCC) screening in patients with hepatitis C virus - advanced chronic liver disease and sustained viral response after oral antiviral treatment. The paper considers multiple parameters and stratifies patients according to the degree of fibrosis quantified by transient elastography, ultrasonography, and serological tests (FIB4, APRI). The authors use complex statistical analysis to identify a subgroup of patients with a very low probability of developing HCC, who could be excluded from screening programs.
In my opinion, some aspects could be corrected/improved:
· In Methods: explanation of all analyzed parameters (e.g., fatty liver index) and the choice of cut-off values for FIB4 and APRI;
· In Results: p. 8 -9, Table 3 albumin > 4g/dL and cholesterol > 150 mg/dL seem to be risk factors for HCC, while in the text is said that “low levels of albumin, cholesterol” were associated with HCC occurrence
· In Discussion: what is the author's explanation that in their cohort alcohol consumption or diabetes mellitus were not risk factors for HCC as in many other cohorts?
Reviewer 2 Report
It is a well-constructed and conducted study, but the authors do not make any reference to the latest Baveno consensus regarding the stiffness value in chronic hepatitis C patients with significant/advanced fibrosis. I think that in table 2, in group 2 liver stiffness values are compatible in some cases with liver cirrhosis. Moreover, the authors specify the possibility that some cirrhotic patients could have been included in Group 1 and 2, due to the wide range of kPa adopted to define F3 patients
The conclusions are logical, but as long as diagnostic accuracy and the optimum cut-off value of both APRI and FIB-4 are not yet very well established in comparison with liver stiffness in chronic hepatitis C patients, I think it is risky not to monitor patients with F3.
Line 35 - References must be written in the MDPI style
Reviewer 3 Report
The study by Ciancio et al. is a retrospective study. The authors excluded 64 (5.4%) patients with hepatocellular carcinoma (HCC) and included 1000 hepatitis C virus-infected patients with liver stiffness measurement (LSM) > 9.5 kPa who underwent antiviral therapy and achieved sustained virological response (SVR). Patients were classified according to their baseline LSM, APRI and FIB-4 scores, which were cut off into 4 groups. Groups 1 and 2 with LSM<14.5 (n=457) and Groups 3 and 4 (n=543) with LSM>14.5 kPa or other signs of cirrhosis. After a median follow-up of 48 months, 71 patients developed an HCC and 60 died (42 due to liver-related causes). The HCC incidence rate (IR) was 1.97/100 patients/year (PY). Multivariate analysis showed that male gender, cirrhosis, oesophagal varices, and groups 2-4 were independent predictors of HCC. However, group 1 (n=324) (LSM 9.5-14.5 without other signs of cirrhosis and FIB-4 < 3.25 and APRI < 1.5) was not related to HCC and its HCC IR was 0.09/100 PY. Patients from Groups 2-4 with a decrease of FIB-4/APRI below 3.25 and 1.5 after SVR, decreased their HCC IR.
The authors concluded that the great majority of F3 patients achieving SVR can be safely suspended from HCC surveillance.
The study has important limitations to be accepted in Biomedicines.
Major comments
The study has important limitations. The authors have recently published (Cancers 2022, 14, 828. https://doi.org/10.3390/cancers14030828) similar results regarding their cirrhotic patients included in Groups 3 and 4 (n=543 out of 575). Therefore, the study will be more interesting if the authors evaluate the HCC IR incidence in patients EXCLUDING CIRRHOTIC PATIENTS such as LSM 9.5-14.5 kPa without other signs of portal hypertension (Groups 1 and 2) and try to identify patients with lower HCC IR according to their variables as LSM, APRI, FIB-4, gender….with their best cut-off (based on diagnostic accuracy, AUROC).
A more interesting perspective is to do an “intention to diagnose analysis” describing excluded patients with HCC before SVR in groups 1 and 2 to avoid selection errors. Please include a table describing differences between excluded and included patients
Please change Tables 1 and 2 with a new one describing differences between those patients who developed (or did not) HCC to identify individual variables related to HCC incidence
Please change the discussion section according to the recommendations.
Around 70% of patients with LSM 9.45-14.5 without other signs of cirrhosis achieving SVR could be safely suspended from HCC. However, it is necessary to validate the results in multicentric prospective studies to make a solid recommendation.
Minor comments
Title:
Please include …..”without cirrhosis”
Abstract:
Include only patients without LSM>14.5 or indirect signs of portal hypertension
Patients and methods
Please, include the previously published group reference regarding cirrhotic patients (https://doi.org/10.3390/cancers14030828)
Please, include the number of Ethics Committee
Please, include the references of all non-invasive diagnostic models and their reference cut-offs
Please, include the definition of clinical and imaging variables to exclude cirrhosis based on a previously published reference
Statistical analysis
-Authors can identify patients with lower HCC IR according to their variables as LSM, APRI, FIB-4, and gender….with their best cut-off (based on diagnostic accuracy, AUROC) to exclude HCC.
-Please, generate the predictive model with the independent variables and their coefficients
Results
Please, describe liver-related causes in groups 1 and 2
Please describe patients with improvement (or not) of LSM/APRI/FIB-4 to identify related variables
Tables and Discussion
See major comments
Reviewer 4 Report
Ciancio, et al. investigated a risk of HCC occurrence after HCV eradication by DAA from a 2-year-prospective study. They found that few patients who showed F3 with low levels of FIB-4 and APRI before therapy developed HCC. They also found that few patients who improved their FIB-4/APRI under 3.25, 1.5, respectively, at SVR achievement did not develop HCC. Then, they concluded that those patients are not necessary for HCC surveillance after HCV eradication by DAA.
Major
1) It is true that vast majority of patients who satisfied their criteria did not develop HCC. However, a few patients developed HCC even in Group 1. It is very important to care these patients and these patients should not be ignored. Please discuss how to monitor these patients. Are there any differences between patients with or without HCC in Group 1?
2) Please describe how to confirm the diagnosis of HCC (enhanced MRI, biopsy, operation…, etc).
3) Please clarify the term “SVR”. Is it SVR 12w or SVR 24w?
4) HCC occurrence generally differ among HCV genotypes. Please show genotypes of each patients.
Minor
1) There are many unnecessary spaces in the whole their manuscript.
Round 2
Reviewer 3 Report
The study persists with important limitations that do not allow the publication in biomedicines.
· The authors considered categorized APRI/FIB4 and LSM as good predictors of HCC incidence before multivariate analysis. However, the authors should identify by univariate and multivariate analysis the differences between patients who developed (or did not) HCC and use these individual variables to predict the HCC incidence
· Please, generate the predictive model with the independent variables and their coefficients with their best cut-off (based on diagnostic accuracy, AUROC) to exclude HCC.
· Finally, the authors do not validate their results. The study does not include an internal group or external validation. Please, consider previously published studies about this subject (http://dx.doi.org/10.1016/j.jhep.2018.07.024) or (https://doi.org/10.1016/j.jhep.2021.11.025)
· The authors used the previously published LSM cutoffs to identify advanced fibrosis (stage F3-4) and cirrhosis (F4) as the same to identify F3 or F4 despite LSM and APRI/FIB-4 have not been validated to discriminate between contiguous fibrosis stages. Authors should use “LSM 9.5-14.5” but not “F3” and LSM >14.5 but not “F4”.
· Finally, include these important limitations in your discussion.
Author Response
A point-by-point response was provided in the attached file.

Reviewer 4 Report
1(1) It is true that vast majority of patients who satisfied their criteria did not develop HCC. However, a few patients developed HCC even in Group 1. It is very important to care these patients and these patients should not be ignored. Please discuss how to monitor these patients. Are there any differences between patients with or without HCC in Group 1? Only one patient in Group 1 developed HCC out of 324 patients (0.3%) 7 months after SVR achievement corresponding to a HCC IR of 0.09/100 PY and a CIR at 12, 36 and 60 months of 0.3%. Baseline characteristics of this patient were not different from those of the remaining 323 patients of Group 1. Considering current recommendations, surveillance of this subset of patients is not cost-effective.
I understand that only a few patients developed in Group 1 and that it is not cost-effective for surveillance of these patients. The title of this article is “WHO SHOULD NOT BE SURVEILLLED FOR HCC DEVELOPMENT…..”, but authors put up a line of defences like “a word of caution is still necessary regarding……” (p15, line 8). I am wondering in which situations authors stand. If authors think it is not cost-effective for surveillance of these patients, authors should describe it in the manuscript.
2(2) Please describe how to confirm the diagnosis of HCC (enhanced MRI, biopsy, operation…, etc): your suggestion has been met, see Patients and Methods, page 8, line 6-9.
OK
3(3) Please clarify the term “SVR”. Is it SVR 12w or SVR 24w? This information has been added, see Patients and Methods page 6 line 25, page 7 line 1.
OK
4(4) HCC occurrence generally differ among HCV genotypes. Please show genotypes of each patients. As reported in Results, page 9, line 22-25 we checked each patient for variables predicting HCC by univariate and multivariate analysis and HCV genotype did not emerge as predictor both in univariate and multivariate model, However, as per your suggestion, we added the information regarding the genotypes of patients developing HCC (see Results, page 9, line 25-26, page 10, line 1-2).
OK
Author Response
A point-by-point response has been provided in the attached file.

Round 3
Reviewer 3 Report
Accept in present form